# The Influence of Thickness on the Tensile Strength of Finnish Birch Veneers under Varying Load Angles

Maximilian Pramreiter [1,*] , Alexander Stadlmann [1], Christian Huber [1], Johannes Konnerth [1] , Peter Halbauer [1], Georg Baumann [2] and Ulrich Müller [1]

1 Department of Material Sciences and Process Engineering, Institute of Wood Technology and Renewable Materials, University of Natural Resources and Life Sciences Vienna, Austria (BOKU), Konrad Lorenz Strasse 24, 3430 Tulln an der Donau, Austria; alexander.stadlmann@boku.ac.at (A.S.); Christian.huber@boku.ac.at (C.H.); johannes.konnerth@boku.ac.at (J.K.); Peter.halbauer@boku.ac.at (P.H.); ulrich.mueller@boku.ac.at (U.M.)
2 Vehicle Safety Institute (VSI), Graz University of Technology, Inffeldgasse 23/I, 8010 Graz, Austria; georg.baumann@tugraz.at
* Correspondence: maximilian.pramreiter@boku.ac.at; Tel.: +43-147-654-89123

**Abstract:** The development of high-performance, veneer-based wood composites is a topic of increasing importance due to the high design flexibility and the comparable mechanical performance to solid wood. Part of this improved mechanical performance can be contributed to the size effect present in wood. Based on previous findings in the literature, this size effect can be either strengthening or weakening. The presented study investigates the influence of thickness and load angle on the tensile strength and tensile stiffness of peeled veneers compared to thin sawn timber. Veneers with thicknesses of 0.5 ± 0.05 mm, 1.0 ± 0.05 mm and 1.5 ± 0.05 mm as well as sawn wood with thicknesses of 1.5 ± 0.1 mm, 3.0 ± 0.1 mm and 5.0 ± 0.1 mm were tested in tension under different load angles (0°, 45° and 90°). The results only partly confirm a size effect for strength parallel to the grain. The strength perpendicular to the grain increased significantly between 0.5 mm and 1.5 mm, with a significant decrease between 1.5 mm and 5.0 mm. The presence of lathe checks diminished the strength perpendicular to the grain of the veneers by about 70% compared to solid wood, partly overshadowing a possible strengthening effect. It was concluded that a transition from a strengthening to a weakening behaviour lies in the range of multiple millimetres, but further investigations are needed to quantify this zone more precisely. The presented results provide a useful basis for the development of veneer-based wood composites with a performance driven layer-thickness.

**Keywords:** birch wood; fibre load angle; mechanical performance; size effect; veneers



## 1. Introduction

The mechanical performance of wood and wood-based products is influenced by a multitude of inherent (e.g., density, grain angle and moisture content) as well as external (e.g., production technology, final dimensions and load case) factors [1]. The final dimensions of the specimen highly influence the overall strength of the material as well as the sensitivity to defects [2]. The general perception in literature states that, with increasing dimensions, the strength of a material decreases [3,4]. This relationship is commonly known as the size effect and is usually described applying Weibull's [5] weakest link theory for brittle materials. According to the theory, the ultimate strength of a material can only be as high as the strength of its weakest link. In regards to wood, the relationship between specimen size (length, width and height/thickness) and mechanical performance has been investigated in different directions (parallel and perpendicular to the grain), in different load cases (tension, compression and bending) and on different size levels (macroscopic and microscopic). The results presented in the literature (see Table 1) partly confirm a size effect for wood, depending on the load case and investigated size scale.

Considering macroscopic specimens of multiple millimetres and centimetres loaded in bending, an increase in size leads to a decrease in strength [6–10]. However, according to Madsen and Buchanan [8] as well as Bohannan [6], the effect of length is superior compared to the width of the samples when loaded in bending. This is supported by results from Chaplin and Nevard [10]. They investigated the influence of defects on bending samples of increasing cross section and did not report a size effect. However, according to Ylinen [9] and Bohannan [6], the cross section very much influences the bending strength. For samples loaded in tension, a decrease in strength with increasing size is reported [11–14]. Contrarily, the results presented in the literature [15–18] for macroscopic samples loaded in compression did not confirm a size effect. While Hu et al. [18] as well as Zauner and Niemz [16] reported a size effect for specimens loaded in compression, Schlotzhauser [17] as well as Mukan Fotsing and Foujet [15] did not find a size effect to be present. In the size literature, microscopic specimens starting at multiple micrometres [19–21] have a reversed relationship, concluding that strength increases with an increase in size. Therefore, the size effect on the strength of wood could be categorized as "strengthening" and "weakening" effects.

This discrepancy was investigated by Buchelt and Pfriem [22]. They investigated the influence of specimen thickness on the tensile strength of thin veneers (0.5 mm) and compared it to solid wood (4 mm) loaded parallel and perpendicular to the grain. According to the results presented, the strength perpendicular to the main fibre direction and the corresponding stiffness of the thin veneers were significantly lower than those of solid wood. However, they found no significant difference in strength parallel to the fibre between veneers and solid wood. They concluded that further investigations are needed to identify the transition zone between the strengthening and the weakening effects. In an earlier study [23], they already confirmed that there was no significant change in mechanical properties for thicknesses below 0.5 mm.

**Table 1.** Chronological overview of the literature investigating the size effect on the mechanical performance of wood and if a size effect was reported (yes) or not (no). (LD = Load direction: Parallel to the grain = ∥ and Perpendicular to the grain = ⊥; size scale: MAC = Macroscopic, MIC = Microscopic; Load cases: TEN = Tension, COM = Compression and BEN = Bending; ↓: strength decreases with increasing size; and ↑: strength increases with increasing size).

| Reference | LD | MAC | MIC | TEN | COM | BEN | Comment |
|---|---|---|---|---|---|---|---|
| Chaplin and Nevard [10] | ∥ | x | | | no | no | Constant length in bending |
| Graf and Egner [24] | ∥ | x | | yes ↓ | | | Increasing cross section |
| Ylinen [9] | ∥ | x | | | | yes ↓ | Constant length, defects and defect free samples |
| Bohannan [6] | ∥ | x | | | | yes ↓ | Increasing length and cross section |
| Schneeweiß [3] | ∥ | x | | | | yes ↓ | Increasing length and cross section |
| Barrett [11] | ⊥ | x | | yes ↓ | | | Theoretical approach |
| Kunesh and Johnson [12] | ∥ | x | | yes ↓ | | | Constant thickness, increasing width |
| Madsen and Buchanan [8] | ∥ | x | | yes ↓ | | yes ↓ | Length effect in bending does not apply in tension |
| Madsen [25] | ∥ | x | | | | yes ↓ | Size effect is best shown as volume effect |
| Madsen [26] | ∥ | x | | | | yes ↓ | Length is superior to depth and width |
| Madsen [2] | ∥ ⊥ | x | | | | | Literature review (book) |
| Aicher and Reinhardt [27] | ∥ | x | | | | | Theoretical approach |

**Table 1.** *Cont.*

| Reference | LD | MAC | MIC | TEN | COM | BEN | Comment |
|---|---|---|---|---|---|---|---|
| Glos and Burger [28] | ∥ | x | | yes ↓ | | | Length effect in tension applies |
| Mukam Fotsing and Foudjet [15] | ∥ | x | | | no | yes ↓ | Hardwood samples |
| Burger and Glos [29] | ∥ | x | | yes ↓ | | | Length effect, no depth or width effect |
| Fonselius [30] | ∥ | x | | | | yes ↓ | Length is superior to depth and no effect of width |
| Clouston et al. [31] | ∥ ⊥ | x | | | | | Theoretical approach to predict strength |
| Pedersen et al. [13] | ⊥ | x | | yes ↓ | | | Loaded in tangential direction |
| Astrup et al. [14] | ⊥ | x | | yes ↓ | | | Loaded in radial direction |
| Biblis [19] | ⊥ | | x | yes ↑ | | | Sliced early and latewood specimens |
| Yu et al. [20] | ∥ | | x | yes ↑ | | | Longitudinal stiffness |
| Buchelt and Pfriem [22] | ∥ ⊥ | x | (x) | yes ↑ no | | | No effect parallel and an increase perpendicular |
| Schneeweiß and Felber [32] | ∥ | x | | | | yes ↓ | Load configuration influences size effect |
| Zauner and Niemz [16] | ∥ | x | | | yes ↓ | | Hourglass specimens, with increasing diameter |
| Živković and Turkulin [33] | ∥ | | x | yes | | | No tendency is described |
| Zhou et al. [7] | ∥ | x | | yes ↓ | yes ↓ | yes ↓ | bending and tension superior to compression |
| Schlotzhauer et al. [17] | ∥ | x | | no | yes ↓↑ | yes ↓ | Compression is species dependent |
| Büyüksarı et al. [21] | ∥ | x | x | yes ↓ | yes ↑ | | Compression strength increases with size |
| Hu et al. [18] | ∥ ⊥ | x | | | yes ↓ | | Different effect on strength and stiffness |

Based on the outlined literature, the presented study aims to further investigate the relationship between the thickness of a specimen and the corresponding tensile strength. Therefore, samples made from birch wood with thicknesses of $0.5 \pm 0.05$ mm, $1.0 \pm 0.05$ mm, $1.5 \pm 0.05$ mm, $3.0 \pm 0.5$ mm and $5.0 \pm 0.5$ mm were tested in tension under different load angles ($0°$, $45°$ and $90°$) until complete failure. The main research questions that the study investigates are the following:

Q1. How does the thickness influence the tensile strength and stiffness of thin birch veneers and solid wood with a thickness of multiple millimetres?

Q2. How do the load angles of $0°$, $45°$ and $90°$ influence the relationship between thickness and strength and stiffness of thin birch veneers?

Q3. Is it possible to accurately quantify a transition from a strengthening to a weakening effect?

## 2. Materials and Methods

### 2.1. Sample Preparation

A total of 495 grade A [34], peeled birch veneers (sourced from Koskisen, Järvelä, Finland) with a format of $25 \times 250$ mm$^2$ and thicknesses of $0.5 \pm 0.05$ mm (V0.5), $1.0 \pm 0.05$ mm (V1.0) and $1.5 \pm 0.05$ (V1.5) mm were prepared from $1000 \times 1000$ mm$^2$ veneer sheets using a

circular saw. In order to investigate the strength and stiffness at the load angles 0°, 45° and 90°, the specimens were cut out under the corresponding angle in relation to the longitudinal axis of the veneer. Out-of-plane fibre deviations were not considered, but it is assumed that the overall fibre angle lies close to the desired load angle. Additionally, 207 clear wood specimens of birch sawn wood with the same sample geometry were prepared out of straight grained, high-quality heartwood boards using a circular saw and subsequently planed to achieve thicknesses of 1.5 ± 0.1 mm (S1.5) 3.0 ± 0.1 mm (S3.0) and 5.0 ± 0.1 mm (S5.0). Only load angles at 0° (parallel to the grain) and 90° (perpendicular to the grain) were investigated for these thicknesses. All samples were stored at standard climate conditions (20 °C ± 2 °C and 65% ± 5% relative humidity) [35] until constant mass was reached. The final thickness of the samples was determined as average of three measurements within the measuring area, with an accuracy of ± 0.01 mm using a digital caliper (Mitutoyo series 500, Kawasaki, Japan). The sample geometry and fibre load angles are further depicted in Figure 1A–D. The final sample number for each configuration can be found in Table 2.

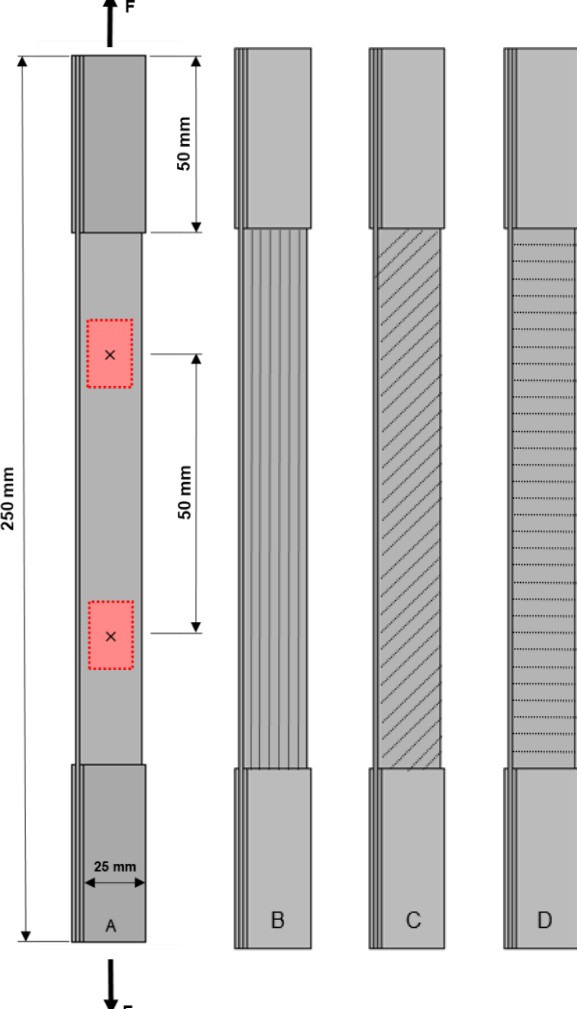

**Figure 1.** (**A**) Sample geometry and schematical depiction of the test setup for strength and stiffness measurements following DIN 52 188 [36] and DIN EN 789 [37], where the gauge length and laser pattern are marked in red and where the reference point for the strain measurement is marked with "x"; (**B**) fibre orientation for 0° load angle measurements ($\sigma_0$ and $E_0$); (**C**) fibre orientation for 45° load angle measurements ($\sigma_{45}$ and $E_{45}$); and (**D**) fibre orientation for 90° load angle measurements ($\sigma_{90}$ and $E_{90}$) (own depiction, not to scale).

**Table 2.** Results of 495 veneer and 207 solid wood samples. (V = veneer, S = Solid wood, t = thickness category, *n* = number of samples tested, σ = strength in the corresponding direction, sd = standard deviation and E = modulus of elasticity in the corresponding direction).

| Group | | 0° | | | 45° | | | 90° | |
|---|---|---|---|---|---|---|---|---|---|
| t (mm) | *n* (–) | σ ± sd (MPa) | E ± sd (GPa) | *n* (–) | σ ± sd (MPa) | E ± sd (GPa) | *n* (–) | σ ± sd (MPa) | E ± sd (GPa) |
| V0.5 | 76 | 121 ± 26 | 13.7 ± 1.6 | 20 | 3.78 ± 0.83 | 0.728 ± 0.243 | 72 | 1.90 ± 0.34 | 0.270 ± 0.629 |
| V1.0 | 74 | 108 ± 30 | 14.4 ± 2.0 | 21 | 4.59 ± 0.64 | 0.785 ± 0.152 | 76 | 2.53 ± 0.43 | 0.327 ± 0.679 |
| V1.5 | 75 | 149 ± 30 | 18.7 ± 3.7 | 18 | 4.22 ± 0.51 | 0.884 ± 0.186 | 63 | 2.94 ± 0.41 | 0.390 ± 0.105 |
| S1.5 | 19 | 108 ± 38 | 14.1 ± 2.1 | - | - | - | 39 | 10.7 ± 1.80 | 0.924 ± 0.145 |
| S3.0 | 13 | 84 ± 21 | 13.9 ± 2.4 | - | - | - | 12 | 6.37 ± 1.32 | 0.996 ± 0.240 |
| S5.0 | 42 | 134 ± 31 | 15.3 ± 3.0 | - | - | - | 82 | 8.45 ± 2.04 | 0.924 ± 0.286 |
| Veneer | 225 | 126 ± 34 | 15.6 ± 3.4 | 59 | 4.20 ± 0.75 | 0.796 ± 0.204 | 211 | 2.44 ± 0.58 | 0.326 ± 0.093 |
| Solid wood | 74 | 119 ± 37 | 14.8 ± 2.7 | - | - | - | 133 | 8.92 ± 2.30 | 0.931 ± 0.250 |

### 2.2. Experimental Characterization

The tensile strength ($\sigma_0$, $\sigma_{45}$ and $\sigma_{90}$) of the samples was determined following DIN 52 188 [36] and modulus of elasticity ($E_0$, $E_{45}$ and $E_{90}$) according to DIN EN 789 [37] using an universal testing machine (Z20 Zwick/Roell, Ulm, Germany) with a load capacity of 20 kN. Usually, these tension specimens are bone shaped in order to achieve failure in the desired area of smallest cross section and to prevent a break in the clamping area [36]. In regard to the thin veneers with perpendicular fibre orientation, milling of the bone shape would have caused an unfeasible rejection rate. Therefore, samples with a uniform cross section were chosen. In order to prevent failure within the clamping area during testing, additional veneers with parallel oriented fibres were glued on both sides using a polyvinylacetat (PVAC) adhesive (Pattex PV/H Express, Düsseldorf, Germany), covering the full clamping area of $25 \times 50$ mm$^2$. The samples were loaded with a pre-force of 20 N. After the pre-force was reached, the samples tested parallel to the fibre were loaded at a constant cross-head speed of 2 mm/min while samples for perpendicular and intermediate loading were loaded at 1 mm/min in order to achieve failure in $90 \pm 30$ s. The elongation was recorded on one side of the samples using a laser extensometer (laserXtens, Zwick/Roell, Ulm, Germany) with a gauge length of 50 mm. The contact-free measuring principle of the laser extensometer prevented pre-damage of the thin samples during setup. The test was stopped after 30% load reduction of the maximum force ($F_{max}$) was reached. The strength was calculated according to DIN 52 188 [36], and stiffness was calculated as a regression curve between 10% and 40% of $F_{max}$ according to DIN EN 789 [37]. The test setup and the centre of the evaluation area of the laser extensometer are further depicted in Figure 1A. Specific strength ($\sigma_{0spec.}$, $\sigma_{45spec.}$ and $\sigma_{90spec.}$) and specific stiffness ($E_{0spec.}$, $E_{45spec.}$ and $E_{90spec.}$) were calculated using the corresponding density of the veneer sheet or wood board, respectively.

### 2.3. Statistics

Data were processed and descriptive statistics were performed using Excel 2016 (Microsoft, Redmond, WA, USA), while one-way ANOVA as well as the post-hoc tests (Games-Howell and Gabriel) were conducted using SPSS 24.0 (IBM SPSS Statistics version 24.0, IBM, New York, NY, USA). In order to compare the different load angles, the effect size ($\omega^2$) was calculated based on Hays [38].

## 3. Results and Discussion

### 3.1. Mechanical Properties

The mean values and the standard deviation obtained from 702 tensile samples are summarized in Table 2 according to the corresponding thickness (V0.5, V1.0, V1.5, S1.5, S3.0 and S5.0) and the respective load angle (0°, 45° and 90°).

The average tensile strength as well as the average tensile modulus decreased significantly overall categories comparing loading parallel and perpendicular to the fibre. This cor-

responds with values found in the literature for loading parallel ($\sigma_{0,mean}$ = ~130–140 MPa and $E_{0,mean}$ = ~13.3–16.2 GPa [39] p. 35) as well as perpendicular ($\sigma_{90,mean}$ = ~3.3–6.3 MPa [40] p. 5–4) to the grain for clear wood specimens. The steep decrease from 0° over 45° to 90° also corresponds with previous findings [41] for veneers as well as solid wood [42,43]. While the mechanical performance in loading parallel to the fibre direction is somewhat similar for veneers ($\sigma_{0,mean}$ = 126 ± 34 MPa and $E_{0,mean}$ = 15.6 ± 3.4 GPa) as well as solid wood ($\sigma_{0,mean}$ = 119 ± 37 MPa and $E_{0,mean}$ = 14.8 ± 2.7 GPa), there is a substantial difference when loaded perpendicular to the fibre. In this case, solid wood samples ($\sigma_{90,mean}$ = 8.92 ± 2.30 MPa and $E_{90,mean}$ = 0.931 ± 0.250 GPa) achieved significantly higher values compared to veneers ($\sigma_{90,mean}$ = 2.44 ± 0.58 MPa and $E_{90,mean}$ = 0.326 ± 0.093 GPa). A possible reason for that will be discussed in the next section.

The average density varied between 483 kg/m$^3$ and 686 kg/m$^3$ over all samples. Veneers with 0.5 mm thickness had a density of 639 ± 12.8 kg/m$^3$, while 1.0-mm-thick veneers exhibited a density of 573 ± 26.4 kg/m$^3$ and 1.5-mm-thick veneers showed the lowest density at 509 ± 10.1 kg/m$^3$. Solid wood samples at 1.5 mm had a density of 637 ± 40.1 kg/m$^3$, 3.0-mm samples averaged 553 ± 10.2 kg/m$^3$ and 5.0-mm-thick samples had a density of 623 ± 28.9 kg/m$^3$. These values are in part significantly lower than the values reported by Sell (650–730 kg/m$^3$) [39], Wagenführ (690–800 kg/m$^3$) [44] or Ross (620 kg/m$^3$) [40]. As density is one of the main factors influencing the mechanical properties of wood [45], this high variation in density between the groups needs to be considered. Therefore, the tensile strengths as well as elastic moduli were also corrected for density in the following section.

### 3.2. Size Effect

In order to investigate the influence of thickness as well as load angle on the mechanical performance, a one-way ANOVA was carried out. The results are summarized in Table 3, and the *p*-values of the appropriate post-hoc tests can be found in the appendix under Table A1.

**Table 3.** The results of one-way ANOVA performed in SPSS 24.0 on the different load angles and different thicknesses as well as effect size ($\omega^2$) based on Hays [38].

| | **0°** | | **45°** | | **90°** | |
|---|---|---|---|---|---|---|
| | **ANOVA** | $\omega^2$ | **ANOVA** | $\omega^2$ | **ANOVA** | $\omega^2$ |
| $\sigma$ | $F_{(5, 293)}$ = 22.173, $p$ = 0.000 | 0.261 | $F_{(3, 56)}$ = 7.325, $p$ = 0.001 | 0.177 | $F_{(5, 338)}$ = 444.116, $p$ = 0.000 | 0.866 |
| E | $F_{(5, 293)}$ = 34.805, $p$ = 0.000 | 0.361 | $F_{(3, 56)}$ = 3.029, $p$ = 0.056 | 0.064 | $F_{(5, 338)}$ = 215.973, $p$ = 0.000 | 0.758 |

Based on the statistical results, a significant difference between the groups for parallel as well as perpendicular loading partly suggests a size effect also reported in the literature [12,20,22,31]. Comparing the load angles, the highest $\omega^2$ was found for loading perpendicular to the fibre, followed by loading parallel and intermediate to the fibre (see Table 3). The high $\omega^2$ for samples loaded perpendicular to the fibre is probably caused by the significant influence of lathe checks, leading to a significant decrease in strength between veneers and solid wood. However, the severity of the lathe checks is in turn influenced by the thickness of the veneer [46]. Therefore, the decrease in mechanical performance perpendicular to the grain can be diminished be an optimized production thickness. This will be discussed in detail later. Figure 2 further illustrates the influence of thickness, load-angle and density on the strength and stiffness. While density was significantly different between the groups, the influence on mechanical performance contradicted the expected behaviour reported in the literature [45], supporting the presence of a strengthening effect in veneers. V1.5 had the lowest average density (509 ± 10.1 kg/m$^3$) but exhibited the highest strength and stiffness parallel to the grain ($\sigma_{0,mean}$ = 126 ± 34 MPa and $E_{0,mean}$ = 15.6 ± 3.4 GPa). This is further illustrated by the specific strength and stiffness in Figure 2 detaching the mechanical performance from the density.

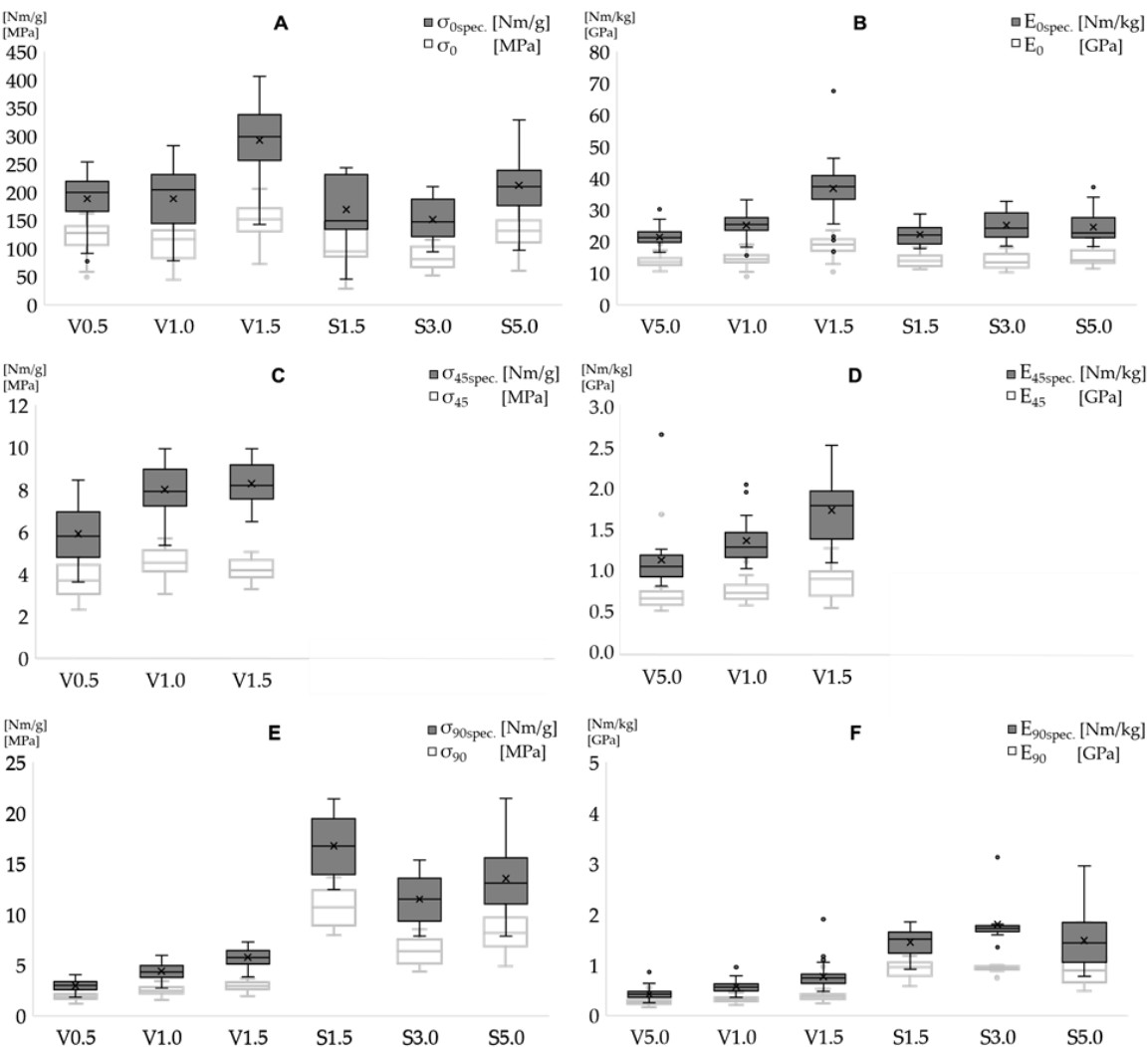

**Figure 2.** Boxplots of strength and stiffness as well as specific strength and stiffness for the respective thickness groups (V0.5, V1.0, V1.5, S1.5, S3.0 and S5.0): (**A**) strength at 0° load angle, (**B**) stiffness at 0° load angle, (**C**) strength at 45° load angle, (**D**) stiffness at 45° load angle, (**E**) strength at 90° load angle and (**F**) stiffness at 90° load angle.

### 3.2.1. Parallel Load Angle

When loaded parallel to the fibre direction (see Figure 2A,B and Table A1), the influence of thickness only partly confirms a size effect. According to the post hoc tests (Table A1), the differences between veneers (V0.5, V1.0 and V1.5) and clear wood (S1.5, S3.0 and S5.0) were not significant for all groups. However, the highest strength as well as stiffness were observed for V1.5 and the increase from V0.5 to V1.5 was significant, as was the decrease from V1.5 to S1.5 and S3.0. This further suggest that, in regards to loading parallel to the fibre, the transition from a strengthening to a weakening behaviour lies somewhere in that region. This is further supported by the density of the samples and illustrated in Figure 2 using the specific strength and specific modulus. A higher density would usually implicate higher strength as well as higher stiffness. However, 1.5-mm-thick veneers had the lowest density (509 ± 10.1 kg/m$^3$) but exhibited the highest strength as well as stiffness. This supports the initial assumption from Buchelt and Pfriemem [22] that thin veneers should be assigned to the category of a strengthening behaviour and further refines a possible transition zone to be somewhere in the range of multiple millimetres. Beside density, other factors influencing strength like fibre orientation [41,47] or moisture content [45] could also explain the increased strength of 1.5-mm-thick veneers. Considering moisture content, all samples were conditioned in the same climate [35] and should therefor exhibit uniform moisture contents. With regards to fibre orientation, some deviation

from the desired angles (0°, 45° and 90°) could be possible. Based on visual inspection, there was no significant difference between the samples. However, as reported in other literature [41,48,49], a slight deviation, especially out-of-plane, from the desired direction already causes significant losses in strength. This could provide a different explanation for the increased strength of the 1.5-mm-thick veneers compared to the clear wood samples, especially for the clear wood samples with the same thickness of 1.5 mm.

### 3.2.2. Perpendicular Load Angle

The influence of thickness when tested perpendicular to the fibre (see Figure 2E,F and Table A1) is significant throughout all categories. According to the results, strength increased significantly from V0.5 up to S1.5, followed by a significant decrease. This suggests that there is a transition from strengthening to weakening behaviour. The strength of veneers was significantly lower compared to solid wood. Comparing V1.5 and S1.5, the average strength of V1.5 was approximately 70% lower than S1.5. A well-known factor decreasing the perpendicular strength of veneers compared to solid wood is the presence of lathe checks [50]. These lathe checks are caused by peeling or slicing of the veneer. The cutting forces in the vicinity of the blade exceed the strength of the wood material and causes the formation of cracks [51]. These cracks significantly decrease the perpendicular strength of the veneer and occur throughout different thicknesses as well as wood species [46,52,53]. According to Bekhta et al. [54], these checks can decrease the strength of veneers by as much as approximately 87%. Therefore, the comparison of veneers and solid wood under perpendicular loading is unsuitable for the identification of a possible transition zone from strengthening to weakening behaviours. According to Palubicki et al. [53], the number of cracks increase and the intact material cross section decreases with increasing veneer thickness for peeled veneer. This would further decrease the strength with increasing thickness as less load-bearing material remains intact and a faster cascading failure occurs. This is in line with the results presented by Purba et al. [55] for 2.1-mm up to 4.0-mm-thick beech and oak veneers as well as by Daoui et al. [56] for 3.0-mm and 5.0-mm-thick beech veneers but contradicts the findings by Buchelt et al. [46]. They investigated sliced birch veneers and found a reversed relationship for thicknesses ranging from 0.3 mm up to 1.2 mm. Beside the cutting technique, other factors influencing lathe checks, e.g., log soaking temperature or cutting speed [57], could also explain this discrepancy in the literature. Therefore, the increase in overall mechanical performance with increasing veneer thickness while not reaching the same levels of strength and stiffness as solid wood could be explained by the lathe checks overshadowing the size effect. Beside lathe checks, wood rays oriented perpendicular to the tangential surface of the peeled veneers could further decrease the perpendicular strength compared to the solid wood samples, especially compared to S1.5 and S5.0, which are a combination of tangential as well as radial loading. As described in the literature [58–62], wood rays not only serve as transportation and storage tissue but also serve as mechanical reinforcement in the radial direction. According to Burgert et al. [60], the perpendicular strength of beech wood samples in the radial direction (approximately 15 MPa) were significantly higher than that in the tangential direction (approximately 7 MPa). Similar findings for solid birch wood were reported by Beery et al. [59]. In the case of peeled veneers, both lathe checks as well as wood rays represent additional weak points under perpendicular loading. Therefore, a size effect perpendicular to grain should be discussed against the background of different production settings in the future.

### 3.2.3. Intermediate Load Angle

The size effect in relation to solid wood for intermediate loading at 45° (see Figure 2C,D) was not investigated due to the absence of solid wood samples. Additionally, there was no significant difference between the thickness groups except for V0.5 and V1.0 concerning strength, which is further reinforced when density is incorporated. A possible reason

for this difference could be a new attribution to the lathe checks which are even more prominent in thinner veneers [46].

*3.3. Possible Benefits of the Size Effect*

Based on the discussed results, the increased strength of 1.5-mm-thick veneers as well as solid wood could provide a solid basis to develop mechanically optimized wood composites with performance-driven layer thickness. In order to cope with the diminished perpendicular strength due to lathe checks, the application of veneers as part of multi-layered composites could provide a technological solution. Optimally, the failure of an ideal adhesive joint is dominated by wood fractures [63]. Therefore, the top layers of a veneer-based composite should be oriented with the checks facing inwards. The applied adhesive would furthermore fill out the lathe checks during the curing process, re-establishing a full, load-bearing cross section in the mechanically significant outer layers.

Combining the optimized layer thickness with a minimized fibre deviation [41–43,64] could significantly improve the mechanical performance of veneer-based wood composites, e.g., laminated veneer lumber (LVL).

Additionally, the results further support the importance of accurate material data for numerical modelling. Recent investigations deal with the implementation of wood into the mobility sector as part of multi-layered composite structures [65,66]. One key factor is the establishment of material cards for numerical modelling based on standardized tests. The presented results show that it is critical to build these material cards not only based on different wood species but also by considering different production technologies of the raw material and semi-finished products.

## 4. Conclusions

Based on the discussed results, the thickness significantly influences the mechanical performance of birch veneers. Parallel and perpendicular strength and stiffness increased with increasing thickness. Therefore, veneer thickness is an essential factor when applied in structural components.

The significant differences in mechanical performance of various thickness groups suggest that a transition from a strengthening to weakening effect is in the area of multiple millimetres. However, further tests need to be conducted to specify this statement, especially to further quantify the discrepancies between loading parallel and perpendicular to the fibre direction.

Lathe checks as well as wood rays decreased the perpendicular tensile strength of veneers by about 70% compared to solid wood of the same thickness. Therefore, the production technology influences the mechanical performance more significantly than a possible size effect. However, as the amount and depth of lathe checks are influenced by veneer thickness and the cutting technique, an indirect size effect can be modulated by the production technology.

The results support the development of mechanically optimized engineered wood products, e.g., laminated veneer lumber with performance-driven layer thickness, and additionally emphasize the importance of accurate material data for the numerical modelling of wood.

**Author Contributions:** Conceptualization, M.P., A.S., C.H. and U.M.; data curation, M.P.; formal analysis, M.P.; funding acquisition, U.M.; investigation, C.H. and P.H.; methodology, M.P. and C.H.; project administration, U.M.; resources, A.S. and P.H.; supervision, J.K. and U.M.; validation, M.P., A.S., J.K. and G.B.; visualization, M.P., J.K., G.B. and U.M.; writing—original draft, M.P.; writing—review and editing, M.P., A.S., J.K., G.B. and U.M. All authors have read and agreed to the published version of the manuscript.

**Funding:** The results presented in this study are part of the research project "Austria Biorefinery Centre Tulln" (ABCT). The financial support by Amt der Niederösterreichischen Landesregierung (K3-F-712/001-2017) and Weitzer Parkett GmbH & CO KG is gratefully acknowledged. Additionally, the authors are thankful for the financial support by the Austrian Research Promotion Agency (FFG, 861421); by the Styrian Business Promotion Agency (SFG, 1.000.054.442); by the Standortagentur Tirol (FFG861421); and from the companies DOKA GmbH, DYNAmore GmbH, EJOT Austria GmbH, Forst-Holz-Papier, Holzcluster Steiermark GmbH, IB STEINER, Lean Management Consulting GmbH, Magna Steyr Fahrzeugtechnik AG & Co KG, MAN Truck & Bus AG, MATTRO Mobility Revolutions GmbH and Volkswagen AG.

**Institutional Review Board Statement:** Not applicable.

**Informed Consent Statement:** Not applicable.

**Acknowledgments:** The results presented in this paper were partly gathered during the creation of the bachelor thesis of Wolfgang Schaunig. The input and time expended during testing of the samples is gratefully acknowledged.

**Conflicts of Interest:** The authors declare no conflict of interest.

## Appendix A

**Table A1.** *p*-values of the post hoc tests for the thicknesses as well as different load angles using strength and stiffness of the samples.

| Group | | 0° | | 45° | | 90° | |
|---|---|---|---|---|---|---|---|
| **(I)** | **(J)** | **σ** | **E** | **σ** | **E** | **σ** | **E** |
| V0.5 | V1.0 | 0.079 | 0.166 | 0.001 | 0.730 | 0.000 | 0.000 |
| | V1.5 | 0.000 | 0.000 | 0.144 | 0.052 | 0.000 | 0.000 |
| | S1.5 | 0.725 | 0.945 | | | 0.000 | 0.000 |
| | S3.0 | 0.000 | 1.000 | | | 0.000 | 0.000 |
| | S5.0 | 0.167 | 0.021 | | | 0.000 | 0.000 |
| V1.0 | V0.5 | 0.079 | 0.166 | 0.001 | 0.730 | 0.000 | 0.000 |
| | V1.5 | 0.000 | 0.000 | 0.253 | 0.322 | 0.000 | 0.001 |
| | S1.5 | 1.000 | 0.997 | | | 0.000 | 0.000 |
| | S3.0 | 0.017 | 0.979 | | | 0.000 | 0.000 |
| | S5.0 | 0.000 | 0.491 | | | 0.000 | 0.000 |
| V1.5 | V0.5 | 0.000 | 0.000 | 0.144 | 0.052 | 0.000 | 0.000 |
| | V1.0 | 0.000 | 0.000 | 0.253 | 0.322 | 0.000 | 0.001 |
| | S1.5 | 0.002 | 0.000 | | | 0.000 | 0.000 |
| | S3.0 | 0.000 | 0.000 | | | 0.000 | 0.000 |
| | S5.0 | 0.146 | 0.000 | | | 0.000 | 0.000 |
| S1.5 | V0.5 | 0.725 | 0.945 | | | 0.000 | 0.000 |
| | V1.0 | 1.000 | 0.997 | | | 0.000 | 0.000 |
| | V1.5 | 0.002 | 0.000 | | | 0.000 | 0.000 |
| | S3.0 | 0.224 | 1.000 | | | 0.000 | 0.919 |
| | S5.0 | 0.110 | 0.529 | | | 0.000 | 1.000 |
| S3.0 | V0.5 | 0.000 | 1.000 | | | 0.000 | 0.000 |
| | V1.0 | 0.017 | 0.979 | | | 0.000 | 0.000 |
| | V1.5 | 0.000 | 0.000 | | | 0.000 | 0.000 |
| | S1.5 | 0.224 | 1.000 | | | 0.000 | 0.919 |
| | S5.0 | 0.000 | 0.526 | | | 0.002 | 0.929 |
| S5.0 | V5.0 | 0.167 | 0.021 | | | 0.000 | 0.000 |
| | V1.0 | 0.000 | 0.491 | | | 0.000 | 0.000 |
| | V1.5 | 0.146 | 0.000 | | | 0.000 | 0.000 |
| | S1.5 | 0.110 | 0.529 | | | 0.000 | 1.000 |
| | S3.0 | 0.000 | 0.526 | | | 0.002 | 0.929 |
| post hoc test | | Games–Howell | | Gabriel | | Games–Howell | |

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
