# Peer review of "The Influence of Thickness on the Tensile Strength of Finnish Birch Veneers under Varying Load Angles"

_forests, doi:10.3390/f12010087_

Round 1

Reviewer 1 Report

The manuscript is well prepared and scientifically interesting. It shows how the size of wooden elements affects their strength properties, confirming the existence of the size effect for wood. However, I have some minor suggestion:

  1. In line 88 sentence change number 495 into words or rewrite the sentence.
  2. “Results and Discussion”, subchapter “Mechanical properties” I would suggest to resign from using min. and max. values in results especially when a table includes mean values and standard deviations only. In „Mechanical properties” the authors describe the density of the samples, however, there is no statistical analysis in terms of the influence of the density on the tested factor. There is also no evidence that the differences in density between the variants are statistically significant. These deficiencies are incomprehensible as the authors themselves emphasize that the density affects the examined factors in an actual way: „As density is one of the main factors influencing the mechanical properties of wood [45], the variation in density be-168 tween the groups needs to be considered.” (line 167-168).
  3. Since the first sentence in "Conclusions" ends with a colon, I would suggest adding bullet points to the conclusions or rewriting the first sentence.

Reviewer 2 Report

Dear Authors,

I see your paper valuable for potential readers due to highly practical tests and useful and applicable results.

I recommend to revise typing errors in the manscript.

(for example: line 219 "Perpendicular loand angl")

With regards!

Reviewer 3 Report

The issue addressed in the paper discusses the influence of thickness on the tensile strength of Finnish birch veneers under varying load angles. In my opinion, an important and interesting topic was proposed, compatible with the scope of the journal. Of course, the topic of the development of high-performance veneer based wood-based composites is a topic of increasing importance due to the high design flexibility and comparable mechanical performance to solid wood.

Such studies are partially analysed in literature. It would be worth presenting the state of the art in a broader way. State of the art should be more related to your research questions.

Generally, in the proposed scope, the paper was prepared correctly. However, I recommend a few corrections to improve the quality of this article:

- to precisely define the research scenario (it is very general); needed to clarify the scope of the study and consequently a clear, step-by-step, simple, synthetic research pattern; yes, the methodology is described, but I recommend more precision, as the reader should know how to repeat a similar analysis on this basis;

- to explain briefly whether there is  need to use, for instance, other methods (first of all, I suggest explaining what are the weaknesses of the proposed method, and what are the strengths, practical advantages; there is a risk of confirming already known, obvious observations - did the Authors notice this?) - please complete point 3;

that is, supplement the summary descriptive analysis.

I also strongly suggest that recommendations for specific, practical, not only general (and not entirely clear) applications of this research shall be provided (please complete point 5).

The language of this paper is relatively correct, however some descriptions would benefit from being more concise. I think it is worth using the help of a native speaker.
